# Service Life Prediction of Painted Renderings Using Maintenance Data through Regression Techniques

André Petersen [1,*], Ana Silva [2] and Marco González [1]

1 Universidade do Vale do Rio dos Sinos, Av. Unisinos 950-Cristo Rei, São Leopoldo 93022-750, Brazil
2 CERIS, Instituto Superior Técnico, University of Lisbon, Av. Rovisco Pais, 10049-001 Lisbon, Portugal
* Correspondence: andrebbpetersen@gmail.com; Tel.: +55-51-99935-1660

**Abstract:** The increase in rehabilitation actions motivated further research in the scope of service life and/or building maintenance. In defining rational strategies for the buildings' maintenance, reliable tools that model their performance over time are needed. This research evaluates the impact of maintenance actions on painted renderings. More specifically, the objective of this research is to adapt and extend an existing method, evaluating new periods of service life, with and without maintenance actions, in order to be able to regionalize and expand the existing results. This research was carried out based on an extensive fieldwork survey of painted renderings of facades in vertical buildings in the city of Porto Alegre, RS, Brazil, at different times (before, during and after the execution of maintenance actions). For renderings without maintenance actions, an average service life of 14 years is obtained. After maintenance actions, especially cleaning, the estimated average service life is 16 years. After maintenance, cleaning and partial repair actions, the estimated average service life is 34 years. For painted surfaces without maintenance actions, an average service life of 10 years is estimated. After cleaning actions, the estimated average service life is 11 years. After maintenance (cleaning, partial repair actions on renderings and repainting), the average service life is 15 years, until the last period of service life that precedes the end of the rendering's life cycle.

**Keywords:** construction; maintenance; service life; life cycle; inspection; degradation

## 1. Introduction

The decrease of new constructions in developed countries and the increase of rehabilitation needs in order to maintain regular conditions of the vast built heritage motivated the need for further research in the scope of service life and maintenance of buildings and their elements [1–10]. Likewise, Caputo et al. [11], Cheng and Ma [12] and Vringer et al. [13] suggest that, in several developed countries, the number of existing constructions is significantly greater than existing projects for new construction. Silva et al. [14] pointed out that the service life of buildings has assumed an important role due to the progressive degradation of the built heritage and the high costs of construction, maintenance and repair. Nevertheless, in 2021, existing gaps are still identified, although some are synthesized by Silva and Brito [15], with the relevant research on service life prediction of elements of the building's envelope.

Predicting and analyzing in advance the future behavior of buildings is more effective and economical than repairing situations of imminent failure of an element [16]. For this, extensive knowledge of each element's behavior is necessary, such as service life, in-service performance, degradation patterns and maintenance operations and their costs throughout the element's life cycle [14,17]. Therefore, the study of the service life and maintenance of facades is essential to aid rational decisions related to the built environment [14,18].

Facade claddings are essential in terms of buildings' performance, acting as the first and most important layer of protection for walls and structures against external environmental degradation agents (weather and pollution, among other factors) [19–23]. The result

of many years of research on this matter have been published [14,24,25], although some predictions had to be estimated without real maintenance data.

The effectiveness of maintenance regimes in buildings is directly related to the accuracy of the service life prediction of systems, elements and components, i.e., an optimal maintenance plan relies on the quality of planning and execution of maintenance actions and operations [17]. Consequently, given its importance, over time, technical standards were created to predict the service life of buildings' components, such as AIJ, BSI and NSF as cited in Shohet and Paciuk [9], the Canadian Standards Association (CSA) [26] and the International Organization for Standardization (ISO) [27]. Silva et al. [14] also refer the guidelines of Norway, Denmark, the Netherlands, New Zealand, and Australia, the American Society for Testing and Materials ([ASTM]) as cited in Silva et al. [14] and the European Organization for Technical Approvals ([EOTA]) as cited in Silva et al. [14]. In Brazil, there is the ABNT NBR 15.575/2021 [28,29], which guides this matter. Furthermore, there are contemporary and older publications. Sjöström [30] introduces the theme, referring to the ASTM E632 from 1982; the simplification of its method through the CSTB (Centre Scientifique et Technique du Batiment) and NSTI (National Swedish Testing Institute); the RILEM TC-60 CSC (Corrosion of Steel in Concrete) together with the CEB (Comité Euro-Internacional du Béton); the study of photovoltaic solar modules from Coulbert in 1983; the Australian standard 1745 from 1975; and the CIB W60 through the study of Blach and Brand as cited in Sjöström [30], including references from 1972, such as the Australian standard CK 24 [30]. Ortega Madrigal et al. [31] comment, among other publications, on the existence of the first specific service life exam, dated 1958 and elaborated on by Legget and Hutcheon. Soronis [32] also suggests the HAPM Component Life Manual, from 1992, and, at an international level, the first version presented in the Netherlands, in 1995, during a meeting of ISO TC59/SC3/W69.

Among other research, in the scope of the service life applied to the buildings' facades, the following publications can be cited: Chai et al. [33], Ferreira et al. [34], Gaspar and Brito [35] and Silva et al. [36–38] for renderings and/or painted surfaces. Directly related, but with a broader scope, Brito et al. [25], Silva and Brito [15] and Silva et al. [14] can also be mentioned. However, although all the mentioned publications described maintenance data, none of them use the data in a direct manner. Probably, this occurs because academic researchers and industry professionals face difficulties in obtaining this kind of information and because it is difficult to simulate in theoretical models the impact of maintenance actions on the evolution of the degradation of elements over time [14,39–41]. Nevertheless, the work of Ferreira et al. [24] assists decision-makers, through their analysis of the impact of maintenance strategies on different buildings' envelope elements, such as facades, windows and roofs. The maintenance model is based on a Petri net formalism and includes degradation, inspection, maintenance and renewal processes.

To fill this gap, this study proposes a method to estimate the service life and therefore the expected life cycle of painted renderings using maintenance information. In this study, the concept of maintenance corresponds to the definition presented in ISO 15686-1:2011 [27], i.e., that is a "combination of all technical and associated administrative actions during the service life to retain a building, or its parts, in a state in which it can perform its required functions". Petersen et al. [18] describe in detail the different actions carried out in the sample analyzed. Succinctly, the following actions are referred in this study: (i) inspection, which corresponds to the action of collecting all the relevant data to evaluate the degradation condition of the facade but does not contribute to improving its condition; (ii) cleaning actions, which correspond to water jet or manual cleaning in order to eliminate superficial and aesthetic anomalies (e.g., stains and superficial dirt); and (iii) partial repair, which corresponds to the correction of anomalies such as cracking or the partial replacement of mortars. With the degradation severity of these claddings in various periods (before a maintenance action and immediately after the intervention, although during its recovery has also been verified), the impact of the maintenance on the overall degradation severity can be measured. For that purpose, a fieldwork survey was carried out in situ on facades

of vertical buildings located in the city of Porto Alegre, in the state of Rio Grande do Sul, Brazil. The sample collected is composed of 18 in-use building facades. In data processing, two condominiums had to be removed once the anomalies were related to the construction process, not being modellable over time.

## 2. Materials and Methods

To apply statistical techniques to model the degradation process of painted renderings and to predict their service life, the degradation condition of these claddings must be quantified. In most of the related studies, the degradation condition of the element under analysis is evaluated through in situ inspections, based on a qualitative and quantitative analysis of the observed defects, their extent and their severity. In this sense, several classification systems are proposed, which classify anomalies according to a scale of discrete variables or bands that range from the most favorable level (without visual degradation) to the least favorable (generalized degradation and/or loss of functionality). The service life prediction used in this paper is based on an empirical model, proposed by Gaspar and Brito [42,43] in which a numerical index called the severity of degradation (*Sw*) is defined, providing an estimation of the overall state of degradation of the inspected facades Equation (1).

This index is obtained by the quotient of the weighted degraded area by the reference area (equivalent to the entire facade with the highest level of degradation severity detected in the mentioned region). The weighted degraded area is given by multiplying the area affected by the different anomalies by a weighting factor related to the severity of each detected defect (*kn*) and by a weighting factor that reflects the relative weight of each anomaly in the overall facade degradation (*ka,n*).

$$Sw = \frac{\sum(An \times kn \times ka,n)}{A \times \sum(kmax)} \tag{1}$$

In this equation, *Sw* represents the severity of degradation, expressing the overall level of degradation, as a percentage; *An*, the area of the facade affected by an anomaly n, in $m^2$; *kn*, the anomaly multiplier factor n, as a function of its degradation level, which may vary from 0 to 4; *ka,n*, the weighting coefficient corresponding to the relative importance of each anomaly; *A*, the total area of the cladding, in $m^2$; and $\sum kmax$ = sum of the weighting factors for the highest level of degradation for each type of defect in cladding with an area equal to *A*.

Regarding the main anomalies that can occur in in-service external claddings, compromising their service life, different authors have established different nomenclatures and types of anomalies to be observed during inspections. This study uses the list of anomalies suggested by Gaspar and Brito [42], Gaspar [44], Prieto et al. [45] and Silva et al. [14,37] for renderings, which are as follows: stains, cracking and loss of adhesion. For painted surfaces, the types used here are the ones that Chai et al. [33], Silva et al. [14] and Prieto et al. [45] suggested, which include chalking defects in addition to the anomalies identified for renderings. In this study, chalking is considered within the detachment group of defects, because the areas obtained are the ones that the maintenance company have measured to be without adherence or already detached with the aim to charge its clients.

According to Gaspar and Brito [43], the weighting coefficients of anomalies can include the condition level of defect (*kn*) (based on its severity) and the relative importance of defect (*ka,n*), all based on the cost of repair and on the probability of promoting the occurrence of new defects as shown in Table 1.

In Table 2, the suggested weights presented are the ones that Chai et al. [33,46] proposed for painted facades.

In this study, the same parameters are used, since they are calibrated and validated for painted renderings [14]. Figure 1 presents an illustrative example of the application of the methodology to a painted rendering analyzed in this study.

**Table 1.** Weighting coefficients ($k_{a,n}$) for rendered facades.

| Degradation Condition | Stains | | Cracking | | Loss of Adhesion | |
|---|---|---|---|---|---|---|
| 1 | $k_{a,n} = 0.12$ | 2.50 €/m² | $k_{a,n} = 0.95$ | 20.50 €/m² | $k_{a,n} = 1.53$ | 33.00 €/m² |
| 2 | $k_{a,n} = 0.53$ | 11.50 €/m² | $k_{a,n} = 0.95$ | 20.50 €/m² | $k_{a,n} = 1.53$ | 33.00 €/m² |
| 3 | $k_{a,n} = 0.53$ | 11.50 €/m² | $k_{a,n} = 1.12$ | 24.00 €/m² | $k_{a,n} = 1.53$ | 33.00 €/m² |
| 4 | $k_{a,n} = 0.53$ | 11.50 €/m² | $k_{a,n} = 1.53$ | 33.00 €/m² | $k_{a,n} = 1.53$ | 33.00 €/m² |

Source: Gaspar and Brito [43].

**Table 2.** Weighting coefficients ($k_{a,n}$) for painted surfaces.

| Defect | Stains/Color Change | Cracking | Chalking | Loss of Adherence |
|---|---|---|---|---|
| $k_{a,n}$ | 0.25 | 1.00 | 1.00 | 1.50 |

Source: Chai et al. [33,46].

| Before the Maintenance Actions | After Cleaning | After Cleaning and Partial Repair of Cracking and of the Not-Adhered Areas | After Cleaning and Partial Repair of Cracking and of the Not-Adhered Areas, and Repainting | Other Information |
|---|---|---|---|---|

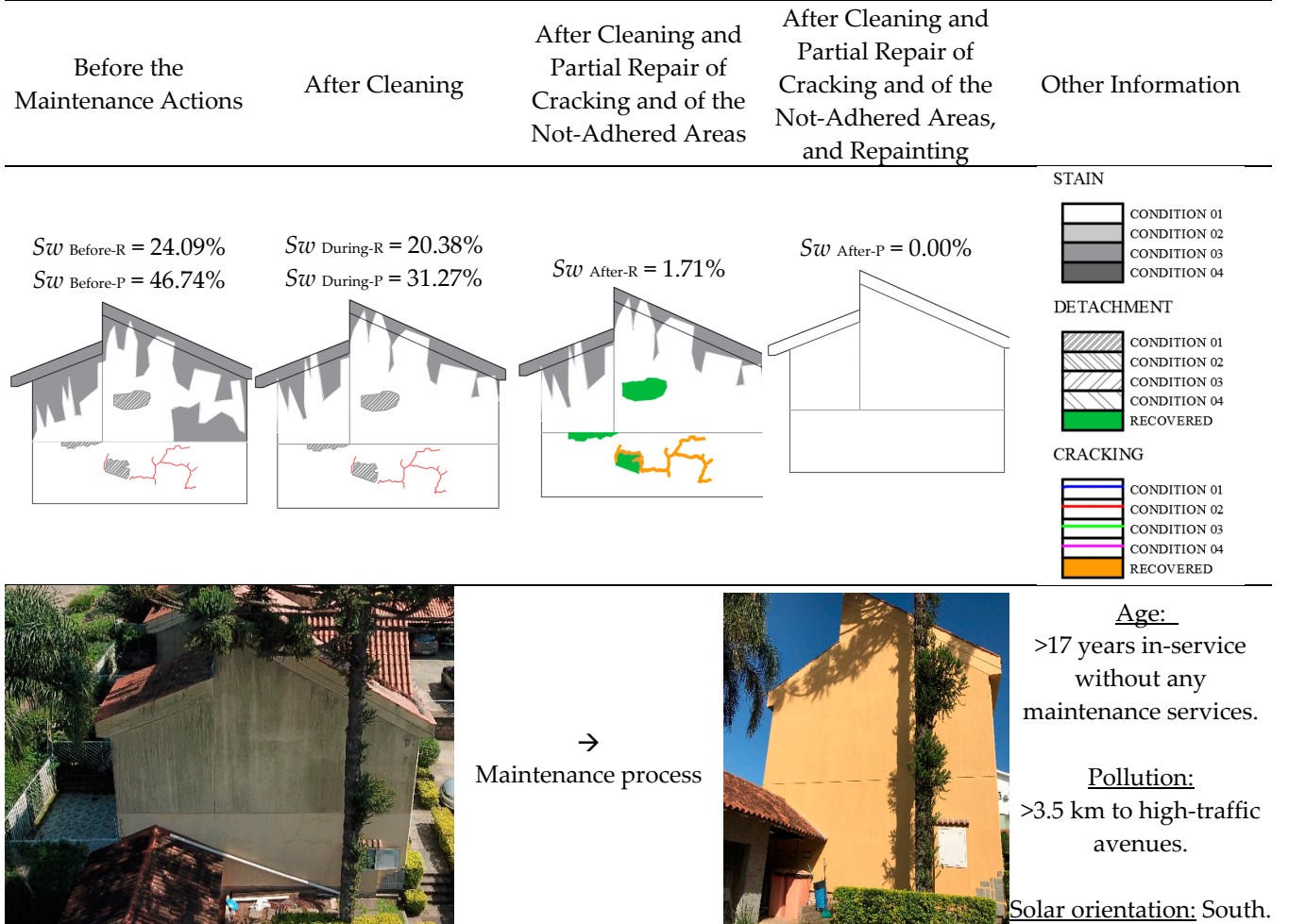

**Figure 1.** Case study of a painted rendering analyzed.

From Tables 1 and 2 and Figure 1, and using Equation (1), the severity of the anomalies is quantified in intervals [14,36,42,44], which allows the definition of the overall condition of the inspected facades and the association of these numerical values to a discrete scale of degradation conditions, as shown in Table 3.

**Table 3.** Definition of degradation levels for renderings and painted surfaces.

| Condition Level | Degradation Level | Renderings | Painted Surfaces |
|:---:|:---:|:---:|:---:|
| A | Level 0 | $S_w \leq 1\%$ | $S_w \leq 1\%$ |
| B | Level 1 | $1\% < S_w \leq 5\%$ | $1\% < S_w \leq 10\%$ |
| C | Level 2 | $5\% < S_w \leq 15\%$ | $10\% < S_w \leq 20\%$ |
| D | Level 3 | $15\% < S_w \leq 30\%$ | $20\% < S_w \leq 40\%$ |
| E | Level 4 | $S_w > 30\%$ | $S_w > 40\%$ |

Source: Adapted from Silva et al. [14].

According to Silva et al. [14], the end of service life of a painted rendered facade occurs in the transition between levels 2 and 3 and/or at 20% of overall degradation. This assumption was defined for the local context (Portugal) but has already shown to be efficient in different countries such as Spain, Iran and in other areas of Brazil, for different types of claddings. Despite the adoption of a method with proven validity in predicting the service life of buildings' facades, the present study takes a step forward, intending to include the impact of maintenance actions (not included in the previous models) in the calculation of the service life of painted renderings through the following sample (Table 4).

**Table 4.** Final samples distribution.

| - | Renderings | Painted Surfaces |
|:---|:---:|:---:|
| Number of data | 79 | 86 |
| Number of in-use constructions/places | 16 | 16 |
| Without previous maintenance | 52 | 54 |
| With previous maintenance | 27 | 32 |
| Dark-colored facades | 25 | 25 |
| Light-colored facades | 54 | 61 |
| Close to pollution sources | 32 | 40 |
| Far from pollution sources | 47 | 46 |
| Obstructed from the sun | 11 | 17 |
| Unobstructed from the sun | 68 | 69 |
| North orientation | 15 | 20 |
| South orientation | 20 | 20 |
| East orientation | 20 | 21 |
| West orientation | 24 | 25 |

| - | Renderings | | Painted surfaces | |
|:---|:---:|:---:|:---:|:---:|
| Age (years) | Partial | Total | Partial | Total |
| Maximum | 21 | 59 | 21 | 59 |
| Average | 10 | 19 | 10 | 20 |
| Minimum | 1 | 1 | 3 | 3 |

| In relation to maintenance—$S_w$ | | Renderings | Painted surfaces |
|:---:|:---:|:---:|:---:|
| Before maintenance services | Maximum | 38.70% | 55.83% |
| | Average | 14.09% | 23.16% |
| | Minimum | 2.15% | 5.90% |
| After cleaning procedure | Maximum | 35.35% | 48.83% |
| | Average | 12.58% | 20.33% |
| | Minimum | 1.94% | 3.40% |
| After cleaning, partial recovery of the renderings and repainting | Maximum | 26.50% | 0.00% |
| | Average | 7.34% | 0.00% |
| | Minimum | 0.45% | 0.00% |

## 3. Life Cycle of Painted Renderings

Previous studies [9,14,36,42,47–49] proposed the adoption of simple regression analysis to create a graphical illustration of the loss of performance of external claddings over time by correlating the age of the facades (in abscissas) and the severity of degradation index (in ordinates). In an advance to the state of knowledge, this study adopts a set of regression analyses to evaluate, sequentially: (i) the proposed extension of the method with the inclusion of maintenance data; (ii) the minimum, average and maximum service life periods, with and without maintenance actions; and (iii) the effects of different maintenance actions in reducing the physical degradation of the inspected facades. The samples collected during the fieldwork survey are used for all the estimations presented in the next section and are divided into simple linear regressions (Section 3.1) and multiple linear regressions (Section 3.2).

### 3.1. Simple Linear Regressions

In this subchapter, the pattern of degradation of painted renderings over time is analyzed. From the collected samples (Table 3), two-dimensional graphs are drawn up through Excel. These graphs have the *Sw* on the *y* axis and the *Age* (period since the last maintenance actions) on the *x* axis. Consequently, when adjusting a linear regression trend from the origin (the equation constant is equal to zero, meaning that at the moment in which the facade is put into use, the degradation is zero), the presented regression equations are obtained. In the sample analyzed, linear degradation patterns presented an adequate goodness-of-fit to the dataset and the parsimony principle was used, adopting a simpler and more effective model in order to model the degradation of painted renderings over time (Figure 2).

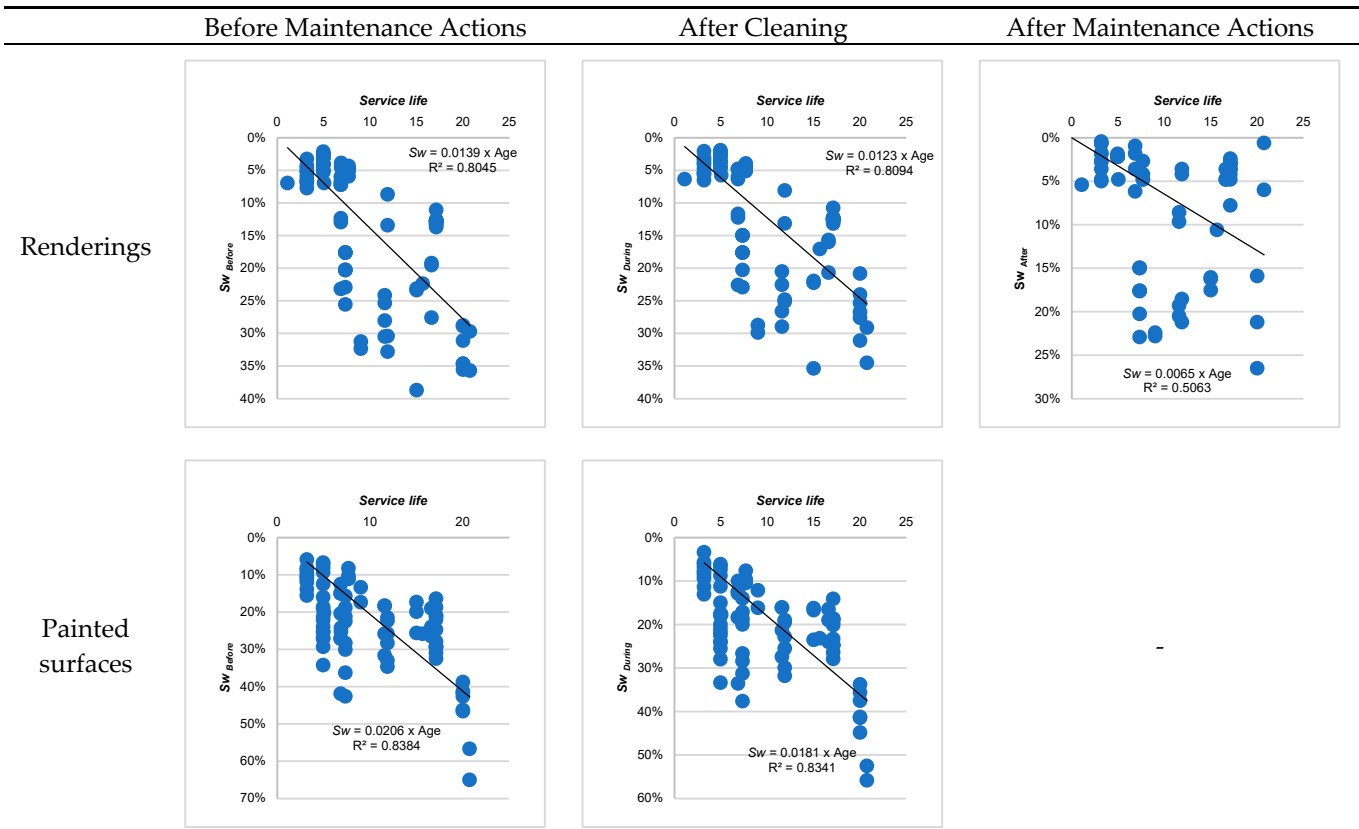

**Figure 2.** Degradation patterns of renderings and painted surfaces over time.

Most of the regression equations present a high determination coefficient ($R^2$), greater than 0.8, which indicates the dimension of the effect of the independent variable, age, on the

dependent variable, severity of degradation. The regression coefficients reveals a positive relationship between the two variables analyzed (age and degradation index), which means that the higher the age, the higher the overall degradation of the I. In Table 5, the regression equations are used to estimate the ages for which a $S_w$ of 20% is obtained, i.e., the age at which the end of the conventional service life is reached.

**Table 5.** Summary of simple linear regression analyses ($Sw$ versus the facades' age).

| Age | Renderings | | | Painted Surfaces | | |
|---|---|---|---|---|---|---|
| Terminology | Service Life $_{(years)}$ | B | $R^2$ | Service Life $_{(years)}$ | B | $R^2$ |
| $S_w$ Before maintenance | 14 | 0.0139 | 0.8045 | 10 | 0.0206 | 0.8384 |
| $S_w$ Before maintenance (average) | 13 | 0.0157 | 0.8932 | 9 | 0.0210 | 0.9097 |
| $S_w$ Cleaning | 16 | 0.0123 | 0.8094 | 11 | 0.0181 | 0.8341 |
| $S_w$ Cleaning (average) | 13 | 0.0159 | 0.8975 | | 0.0185 | 0.9132 |
| $S_w$ After maintenance | 31 | 0.0065 | 0.5063 | - | - | - |
| $S_w$ After maintenance (average) | 27 | 0.0073 | 0.6093 | - | - | - |

The results obtained provide very useful information. Existing studies adopt theoretical assumptions about the impact of maintenance actions on element degradation, but none use historical data or real field data in this analysis. In this sense, the present study allows for validating the theoretical assumptions assumed in the existing literature, since the effect of different types of routine maintenance is directly considered on the service life predictions of painted renderings.

In addition to the information presented in Table 5, other independent variables are tested. In summary, global degradation indexes are identified according to binary frameworks such as the color of the coating; the distance to a source of pollution (distance from high-traffic avenues); the existence of shadow related to the surrounding environment (trees or other constructions); and the predominant solar orientation. Table 6 presents the regression equations and the results obtained.

The results reveal that cleaning actions, due to their general nature and uniform application, lead to a reduction in the overall degradation condition in an almost uniform way. On the other hand, given that stains have a lower severity when compared to cracks and/or loss of adherence, cleaning actions present relatively small importance from the perspective of reducing the overall degradation index. However, this maintenance action is extremely relevant since it promotes an increase in the durability of the facade, since, in theory, it reduces the action of different degradation agents that act together, such as moisture and pollution, for example; further, it has a relevant impact in improving the aesthetic appearance of the facade, increasing the real estate value of the asset.

Regarding the results obtained for the partial repair of renderings, the results reveal the difficulty of obtaining a trend line with a higher determination coefficient ($R^2$). Predicting the impact of localized repairs on the service life of renderings is a challenging task, since the actions carried out are not uniform (they can vary from the simple filling of small cracks to the partial replacement of the render), which implies a significant variation in the global degradation index after maintenance actions, and it is not easy to obtain a value that is applicable to other facades of the same building or of other buildings.

**Table 6.** Summary of simple linear regression analyses ($Ages/S_w$ versus binary frameworks).

| Linear Pattern ($Sw$ = 20%) | Renderings | | | | | | | | | Painted Surfaces | | | | | |
|---|---|---|---|---|---|---|---|---|---|---|---|---|---|---|---|
| | Before Maintenance Services | | | After Cleaning Procedures | | | After Maintenance Services | | | Before Maintenance Services | | | After Cleaning Procedures | | |
| Terminology | Service life | B | $R^2$ | Service life | B | $R^2$ | Service life | B | $R^2$ | Service life | B | $R^2$ | Service life | B | $R^2$ |
| Dark-colored facades | 12 | 0.1660 | 0.8662 | 13 | 0.014 | 0.8828 | 29 | 0.007 | 0.4254 | 8 | 0.0259 | 0.8580 | 9 | 0.022 | 0.8476 |
| Light-colored facades | 15 | 0.0130 | 0.7671 | 17 | 0.011 | 0.7731 | 31 | 0.006 | 0.5337 | 11 | 0.0189 | 0.8572 | 12 | 0.016 | 0.8585 |
| Close to pollution sources (<3 km) | 11 | 0.0185 | 0.9038 | 12 | 0.016 | 0.9023 | 22 | 0.009 | 0.7026 | 9 | 0.0221 | 0.9276 | 10 | 0.02 | 0.9400 |
| Far from pollution sources (>3 km) | 20 | 0.0101 | 0.7538 | 22 | 0.009 | 0.7865 | 45 | 0.004 | 0.3594 | 11 | 0.0182 | 0.7535 | 12 | 0.016 | 0.7316 |
| Obstructed from the sun | 14 | 0.0139 | 0.8012 | 16 | 0.012 | 0.7909 | 29 | 0.006 | 0.4416 | 10 | 0.0194 | 0.9480 | 11 | 0.018 | 0.9315 |
| Unobstructed from the sun | 15 | 0.0136 | 0.7778 | | 0.012 | 0.7936 | 39 | 0.005 | 0.5328 | | 0.0207 | 0.8004 | | 0.018 | 0.7980 |
| North orientation | 16 | 0.0123 | 0.8615 | 18 | 0.011 | 0.8818 | 41 | 0.004 | 0.4775 | 10 | 0.0192 | 0.9205 | 11 | 0.018 | 0.9043 |
| South orientation | 11 | 0.0174 | 0.7836 | 13 | 0.015 | 0.7896 | 26 | 0.007 | 0.4810 | 9 | 0.0214 | 0.8892 | | 0.018 | 0.8917 |
| East orientation | 16 | 0.0126 | 0.8448 | 18 | 0.011 | 0.8528 | 29 | 0.007 | 0.5998 | 10 | 0.0205 | 0.7900 | | 0.018 | 0.7718 |
| West orientation | | 0.0122 | 0.7785 | | 0.011 | 0.7828 | 31 | 0.006 | 0.5394 | | 0.0199 | 0.7715 | | 0.018 | 0.7728 |

### 3.2. Multiple Linear Regression

In this subchapter, the service life of painted renderings is estimated considering the effect of multiple variables. This diligence is intended to describe the variability of the severity of degradation not explained by simple regression, since the variables tend to act synergically, influencing the overall degradation conditions of the claddings. In this situation, the degradation patterns and the periods of service life obtained are more likely to represent reality. Tables 7 and 8 show the multiple linear regression model, created through SPSS software and using the stepwise method (criteria: probability of F to be inserted ≤0.050, probability of F to be removed ≥0.100).

**Table 7.** Summary of the models obtained through multiple linear regression.

| Facade | Model | r | $R^2$ | Adjusted R Square | Std. Error of the Estimate |
|---|---|---|---|---|---|
| | 1 | 0.930 | 0.865 | 0.860 | 0.06518 |
| | 2 | 0.961 | 0.923 | 0.918 | 0.04998 |
| Renderings | 3 | 0.932 | 0.868 | 0.863 | 0.05781 |
| | 4 | 0.959 | 0.919 | 0.913 | 0.04597 |
| | 5 | 0.755 | 0.570 | 0.559 | 0.06867 |
| | 6 | 0.947 | 0.897 | 0.894 | 0.03362 |
| | 7 | 0.942 | 0.888 | 0.884 | 0.08752 |
| Painted surfaces | 8 | 0.886 | 0.785 | 0.775 | 0.12165 |
| | 9 | 0.940 | 0.883 | 0.879 | 0.07903 |
| | 10 | 0.885 | 0.783 | 0.775 | 0.10783 |

**Table 8.** Coefficients obtained through multiple linear regression.

| - | Model | Variables | Unstandardized Coefficients | | Standardized Coefficients | t | Sig. | Collinearity Statistics | |
|---|---|---|---|---|---|---|---|---|---|
| | | | B | Std. Error | Beta | | | Tolerance | VIF |
| Renderings before maintenance | 1 | Partial age | 0.010 | 0.001 | 0.664 | 12.368 | 0.000 | 0.615 | 1.626 |
| | | Pollution | 0.072 | 0.014 | 0.262 | 5.068 | 0.000 | 0.661 | 1.512 |
| | | South | 0.056 | 0.017 | 0.163 | 3.375 | 0.001 | 0.758 | 1.319 |
| | 2 | Total age | 0.005 | 0.000 | 0.690 | 14.459 | 0.000 | 0.457 | 2.188 |
| | | South | 0.056 | 0.013 | 0.161 | 4.327 | 0.000 | 0.755 | 1.325 |
| | | Shadow | 0.065 | 0.016 | 0.138 | 3.920 | 0.000 | 0.835 | 1.198 |
| | | Pollution | 0.036 | 0.012 | 0.132 | 3.080 | 0.003 | 0.567 | 1.764 |
| | | Color | 0.023 | 0.011 | 0.075 | 2.184 | 0.032 | 0.885 | 1.130 |
| Renderings after cleaning actions | 3 | Partial age | 0.009 | 0.001 | 0.668 | 12.583 | 0.000 | 0.615 | 1.626 |
| | | Pollution | 0.064 | 0.013 | 0.262 | 5.124 | 0.000 | 0.661 | 1.512 |
| | | South | 0.049 | 0.015 | 0.158 | 3.307 | 0.001 | 0.758 | 1.319 |
| | 4 | Total age | 0.004 | 0.000 | 0.683 | 13.949 | 0.000 | 0.457 | 2.188 |
| | | South | 0.049 | 0.012 | 0.159 | 4.162 | 0.000 | 0.755 | 1.325 |
| | | Shadow | 0.059 | 0.015 | 0.141 | 3.899 | 0.000 | 0.835 | 1.198 |
| | | Pollution | 0.034 | 0.011 | 0.137 | 3.107 | 0.003 | 0.567 | 1.764 |
| | | Color | 0.021 | 0.010 | 0.077 | 2.183 | 0.032 | 0.885 | 1.130 |
| Renderings after maintenance | 5 | Partial age | 0.005 | 0.001 | 0.553 | 6.118 | 0.000 | 0.684 | 1.461 |
| | | Pollution | 0.047 | 0.015 | 0.290 | 3.206 | 0.002 | 0.684 | 1.461 |
| | 6 | Total age | 0.004 | 0.000 | 0.990 | 24.802 | 0.000 | 0.841 | 1.188 |
| | | North | −0.029 | 0.009 | −0.124 | −3.111 | 0.003 | 0.841 | 1.188 |

**Table 8.** *Cont.*

| - | Model | Variables | Unstandardized Coefficients | | Standardized Coefficients | t | Sig. | Collinearity Statistics | |
|---|---|---|---|---|---|---|---|---|---|
| | | | B | Std. Error | Beta | | | Tolerance | VIF |
| Painted surfaces before maintenance | 7 | Partial age | 0.017 | 0.001 | 0.767 | 15.795 | 0.000 | 0.574 | 1.743 |
| | | Color | 0.093 | 0.019 | 0.195 | 4.946 | 0.000 | 0.871 | 1.148 |
| | | Pollution | 0.049 | 0.018 | 0.130 | 2.777 | 0.007 | 0.615 | 1.625 |
| | 8 | Total age | 0.006 | 0.001 | 0.589 | 8.974 | 0.000 | 0.608 | 1.646 |
| | | Color | 0.139 | 0.025 | 0.293 | 5.474 | 0.000 | 0.916 | 1.092 |
| | | Shadow | 0.107 | 0.033 | 0.185 | 3.245 | 0.002 | 0.805 | 1.242 |
| | | South | 0.071 | 0.031 | 0.134 | 2.290 | 0.025 | 0.767 | 1.303 |
| Painted surfaces after cleaning actions | 9 | Partial age | 0.015 | 0.001 | 0.780 | 15.760 | 0.000 | 0.574 | 1.743 |
| | | Color | 0.089 | 0.017 | 0.210 | 5.233 | 0.000 | 0.871 | 1.148 |
| | | Pollution | 0.032 | 0.016 | 0.097 | 2.026 | 0.046 | 0.615 | 1.625 |
| | 10 | Total age | 0.006 | 0.001 | 0.655 | 11.081 | 0.000 | 0.750 | 1.334 |
| | | Color | 0.131 | 0.023 | 0.310 | 5.803 | 0.000 | 0.918 | 1.089 |
| | | Shadow | 0.090 | 0.029 | 0.176 | 3.095 | 0.003 | 0.808 | 1.238 |

## 4. Discussion of the Results

The results obtained allow us to draw the following conclusions, considering the painted renderings' life cycle:

- Physical degradation of buildings over time is normal and expected. The degradation patterns proposed by ABNT [28], de Flores-Colen and Brito [50] and Gaspar [44], as well as those in this research, demonstrate that, even with maintenance actions, a residual decrease in facades' performance over time is still observed. Some authors refer to the fact that there is an accelerated degradation in the initial phase, which tends to stabilize and accelerates again at the end of the facade's life. Madureira et al. [51] suggested that renderings and painted facades require continuous inspections and maintenance actions to maintain an adequate level of performance.

- The intervening variables in the degradation of the inspected facades were (besides the age) (i) the facades' color; (ii) the distance from a source of pollution; (iii) the existence of shading; and (iv) the solar orientation, especially the one facing south. These conclusions are similar to those obtained by Gaspar [44] and Silva et al. [14,36,37]. However, some variables could not be identified in this research, such as the type of materials applied in renderings or in painted surfaces, since the coatings are inspected in situ several years after application.

- Different authors, such as Lavy and Shohet [52], Moubray [53], Shohet and Paciuk [9,47] and Shohet et al. [48,49] already suggested that linear degradation patterns are adequate and valid to describe the performance of components over time Further:

  ○ Gaspar [44], Gaspar and Brito [42] and Silva et al. [36] also found linear degradation patterns for renderings. Similarly, Silva et al. [37] also adopted a linear equation to model the service life of renderings, even though they followed the analysis with neural networks.

  ○ Chai et al. [33] discussed the application of polynomial and linear trend lines for the service life prediction of painted surfaces.

- The inspections must be carried out every 5 years, to monitor the degradation of the painted renderings. During this period, the degradation condition of these coatings can be monitored in a preventive way, contemporaneous with the likely periods of intervention, as well as at times when corrective prognostics will probably be suggested.

  ○ However, there are shorter and more demanding periods, according to Flores-Colen and Brito [50] and Madureira et al. [51]. On the other hand,

Sá et al. [54,55] point out a better cost–benefit ratio for everyone involved, as they suggest preliminary inspections every 15 or 24 months and detailed inspections between 5 and 10 years.

- Regarding the average service life obtained for renderings without maintenance actions:
  - Is 14 years, ranging from 11 to 20 years, using simple linear regressions;
  - Is 14 years, ranging from 7 to 20 years, using multiple linear regressions;
  - The estimated service life periods obtained in this research are similar to those found by Gaspar and Brito [35,42], Silva and Brito [15] and Silva et al. [14,18,36]. The differences observed among studies are due to the application of different materials and/or in the ways of designing and building and the active degradation agents, among other factors. For example, Afzali and Hamzehloo [56] obtained values lower than those found in this research, which, in turn, on average, are lower than those found in the literature.
  - Regarding the ABNT [28] guidelines, the design service life for this and other types of claddings (ceramics and stones) must be at least 20 years. According to this standard, this period is linked to carrying out maintenance actions on the facade based on other technical and normative guidelines. Therefore, the results obtained during the fieldwork survey on the maintenance actions carried out in painted renderings reveal that the standard is fulfilled, even though users should be informed of the possible need to intervene in shorter periods so that the minimum period of performance is effectively reached.
- The average service life obtained for renderings after cleaning actions:
  - Is 16 years, ranging from 12 to 22 years, using simple linear regressions.
  - Is 16 years, ranging from 10 to 22 years, using multiple linear regressions.
- The average service life obtained for renderings after maintenance actions:
  - Is 34 years, ranging from 22 to 45 years, using simple linear regressions.
  - Is 35 years, ranging from 31 to 40 years, using multiple linear regressions.
- Therefore, based on the results obtained from the sample collected, the expected performance of renderings through their life cycle can be illustrated as shown in Figure 3.

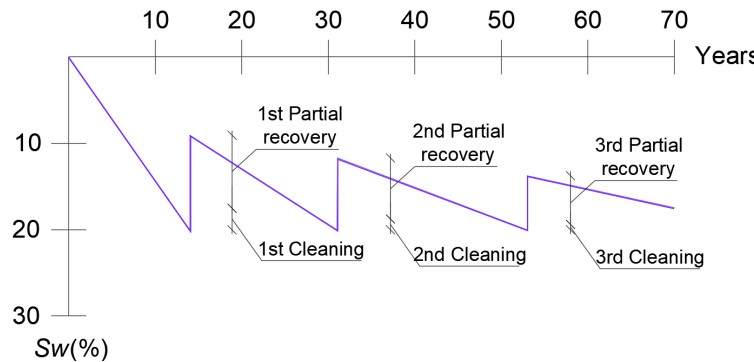

**Figure 3.** Service life and life cycle obtained for renderings.

- The average service life of painted surfaces without maintenance actions;
  - Is 10 years, ranging from 8 to 11 years, using simple linear regressions.
  - Is 8 years, ranging from 3 to 12 years, using multiple linear regressions.
- The estimated service life periods in this research are similar to those found by Chai et al. [33], Silva and Brito [15] and Silva et al. [14,15].
- According to the ABNT [28] guidelines, the design service life for painted surfaces must be at least 8 years. According to this standard, this period is associated with carrying out maintenance actions on the facade based on other technical and normative

guidelines. In the sample analyzed, this normative guideline has been accomplished, but the users should be made aware of the need for early intervention in case unacceptable levels of degradation are reached.

- The average service life of painted surfaces after cleaning actions:
  - Is 11 years, ranging from 9 to 12 years, using simple linear regressions.
  - Is 9 years, ranging from 5 to 13 years, using multiple linear regressions.

- The average service life of painted surfaces after maintenance actions:
  - Is around 11 years, ranging from 5 to 13 years until the last period of service life that precedes the end of its life cycle. This is because, after repainting the facades, a global degradation index equal to 0% is assumed for the full range of interventions.

- Therefore, based on the results obtained from the sample analyzed, the expected performance of painted surfaces through their life cycle can be illustrated as shown in Figure 4.

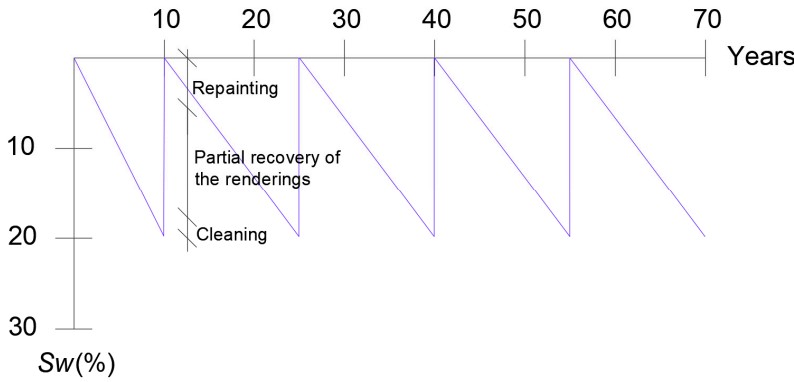

**Figure 4.** Service life and life cycle obtained for painted surfaces.

- More details of the maintenance services that were performed can be found in Petersen et al. [18].

## 5. Conclusions

A culture of thinking about the construction process limited to the design and execution stages was adopted for many years, neglecting subsequent phases. The maintenance phase of buildings is responsible for a significant part of the life cycle costs of buildings, and it is necessary to have reliable data on the behavior and degradation pace of the components, their maintenance needs and the impact of maintenance activities on the buildings' durability. Constructions are built to meet the users' requirements for many years, and, over that time, they must be monitored to present adequate conditions for their intended function, adopting maintenance procedures to avoid and mitigate the action of degradation agents and use conditions, which change their original properties. The physical degradation of buildings over time is normal and unavoidable. The degradation patterns presented in this research reveal that, even with maintenance actions, there is a residual decrease in the facades' performance over time.

Since maintenance has a decisive influence on the life cycle of constructions, it should not be done in an improvised, sporadic or casual way, specifically, without the adoption of technical and rational criteria. Maintenance actions should be considered as a programmable and technical service and as an investment in safeguarding the property's value.

Once it has been validated that the assumption that the application of service life prediction methods still does not consider the maintenance variable directly, it is possible to assume that constructive diagnoses and prognoses, currently, may lack precision in their methods and, consequently, in their conclusions. Such a situation can wrongly impute responsibilities to the interveners regarding the cause and origin of the found anomalies,

based on mistaken cultural assumptions that the constructions are very expensive and, consequently, should have a longer service life without the need for short- and medium-term maintenance costs.

In other words, this study allowed us to identify the differences in the degradation process due to the characteristics of the buildings, the conditions of environmental exposure and, mainly, the impact that maintenance services have on the performance of the painted renderings. This last variable intends to fill a considerable gap in knowledge regarding the impact of maintenance actions on the natural ageing process of constructions. Therefore, whether in terms of quantifying the effective improvement caused by maintenance actions or in terms of service life prediction, a singular analysis was carried out, which allowed obtaining new information to satisfactorily characterize the studied coatings.

The sample analyzed and the models proposed reveal that the adoption of maintenance actions and routine interventions promote the durability of painted renderings over their life cycle, extending their service life, although not infinitely.

As specific conclusions for each coating studied, the service life periods obtained can be summarized. For renderings without maintenance actions, the estimated average service life is 14 years, ranging from 11 to 20 years. After maintenance actions, especially cleaning, the estimated average service life is 16 years, ranging from 12 to 22 years. After maintenance, cleaning and partial repair actions (crack sealing and/or partial repair of debonded regions), the estimated average service life is 34 years, ranging from 22 to 45 years. Regarding the average life cycle of renderings, without maintenance actions, the period of 34 years is obtained, ranging from 27 to 40 years. After maintenance actions, especially cleaning, the average life cycle obtained is 42 years, ranging from 35 to 50 years. After maintenance, cleaning and partial repair actions, the estimated average life cycle is 61 years, ranging from 51 to 71 years.

For painted surfaces without maintenance actions, the estimated average service life is 10 years, ranging from 8 to 11 years. After cleaning actions, the estimated average service life is 11 years, ranging from 9 to 12 years. After maintenance, cleaning, partial repair actions on renderings and repainting actions, the average service life is estimated at approximately 15 years, ranging from 10 to 20 years until the last period of service life that precedes the end of the rendering's life cycle.

This study promotes a significant advance in knowledge in the maintenance area, whether for academics or for the industry, providing a methodology to assess the impact of maintenance actions on building elements. In future, some actions seems fundamental to improve the knowledge of service providers that carry out pathology investigations, deepening the basic rules on this subject. For example, this includes incentives for the creation of new, more accessible or simplified tools, such as *Sw*, so that diagnoses and prognoses have a minimal and technical criterion to be followed. Consequently, this would lead to conclusions and referrals that are more accurate and less sensory. Additionally, an establishment of periodic inspections is suggested, first in a preliminary manner and later in a more detailed manner if necessary, paying attention to the average performance periods obtained for each type of system, element or component that will be analyzed. This would allow for constant monitoring with preventive and/or corrective measures.

**Author Contributions:** A.P. conceived the paper outline and wrote the first draft; A.S. gave academic guidance to this research work and revised the first draft; M.G. revised the conceptualization of the method and reviewed the final manuscript. All authors contributed to every part of the research described in this paper. All authors have read and agreed to the published version of the manuscript.

**Funding:** This research received no external funding.

**Data Availability Statement:** Not applicable.

**Conflicts of Interest:** The authors declare no conflict of interest.

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
