# Peer review of "Service Life Prediction of Painted Renderings Using Maintenance Data through Regression Techniques"

_buildings, doi:10.3390/buildings13030785_

Round 1

Reviewer 1 Report

This paper proposes a method to determine the service life of painted renderings using maintenance information.

From this reviewer point of view, the topic of the manuscript is important and relevant for the section “Building Materials, and Repair & Renovation” of the Journal Buildings. Moreover, the overall methodology is quite adequate regarding the main goals of the paper, and the main findings are reasonably well supported by the obtained results. 

This reviewer suggests the paper to be accepted for publication after minor revision:

1. All variables must be in italic (including the main text); 

2. English should be revised. 

Author Response

Comments and Suggestions for Authors

This paper proposes a method to determine the service life of painted renderings using maintenance information.

From this reviewer point of view, the topic of the manuscript is important and relevant for the section “Building Materials, and Repair & Renovation” of the Journal Buildings. Moreover, the overall methodology is quite adequate regarding the main goals of the paper, and the main findings are reasonably well supported by the obtained results. 

The authors would like to thank the reviewer for the careful analysis of the paper and for the comments that allowed us to improve the document.

This reviewer suggests the paper to be accepted for publication after minor revision:

  1. All variables must be in italic (including the main text);

Done as suggested.

  1. English should be revised.

Done as suggested.

Reviewer 2 Report

The paper is interesting because it analyses and discusses some questions about the service life prediction of painted renderings using maintenance data. But in the text other types of surface finish appear in buildings that are considered. Its content fits the themes of Buildings magazine and it is recommended that it can be accepted after major revision.

There are some issues that must be taken in attention by authors in a revised version:

It is necessary to clarify and define in advance, preferably in the introductory chapter, some basic assumptions for the study that condition the entire work. It should be clearly defined what maintenance means and also the concepts of without maintenance action / maintenance action / cleaning action / etc. and describe the characteristics of the works that are considered in each of these procedures. For example, how is cleaning action different from maintenance action?

Throughout the text, several references are made to different types of finishes: renderings, painted surfaces, facade claddings, painted renderings, external claddings, etc. It is necessary to unify the nomenclature throughout the text. For example, in the abstract, renderings and painted surfaces are mentioned and data on average service life is presented. It is not possible to discern the differences. Is it monomass mortars in the case of renderings?

It is also necessary to clearly describe what type of buildings were studied. Line 85 - “The sample collected is composed of 18 in-use buildings (some with more than one tower or residence). In data processing, two condominiums (…)”. Tower, residence, condominium are different concepts.

Table 1 and 2. Rendered facades, painted facades… The rendered facades are not painted? Table 3 brings together the 2 concepts

Figure 2. The adjustment of the linear regression does not seem very enlightening and the margin of error is very significant. Is it possible to adjust better and draw more evident conclusions? Or be critical of the results obtained?

There is self-citation by one of the authors of the article throughout the text. 12 references to “Silva et al.” were identified.

Author Response

Comments and Suggestions for Authors

The paper is interesting because it analyses and discusses some questions about the service life prediction of painted renderings using maintenance data. But in the text other types of surface finish appear in buildings that are considered. Its content fits the themes of Buildings magazine and it is recommended that it can be accepted after major revision.

The authors would like to thank the reviewer for the careful analysis of the paper and for the comments that allowed us to improve the document.

There are some issues that must be taken in attention by authors in a revised version:

It is necessary to clarify and define in advance, preferably in the introductory chapter, some basic assumptions for the study that condition the entire work. It should be clearly defined what maintenance means and also the concepts of without maintenance action / maintenance action / cleaning action / etc. and describe the characteristics of the works that are considered in each of these procedures. For example, how is cleaning action different from maintenance action?

Done as suggested. In the revised version of the manuscript, the concepts are defined, clarifying the assumptions of the study.

Throughout the text, several references are made to different types of finishes: renderings, painted surfaces, facade claddings, painted renderings, external claddings, etc. It is necessary to unify the nomenclature throughout the text. For example, in the abstract, renderings and painted surfaces are mentioned and data on average service life is presented. It is not possible to discern the differences. Is it monomass mortars in the case of renderings?

Done as suggested. The facades analysed in this study are painted renderings. The goal of this study is to use the Sw equation, and the premisses for each type of cladding, already validated by different studies published in several journals. After regionalizing the referred degradation model, and after considering the different conditions for construction, this study added a new dimension and a relevant variable that had never been used directly before. Since this study is based on the visual inspection of painted renderings several years after application, it is not possible to accurately characterise the type of mortar or paint that were applied.

It is also necessary to clearly describe what type of buildings were studied. Line 85 - “The sample collected is composed of 18 in-use buildings (some with more than one tower or residence). In data processing, two condominiums (…)”. Tower, residence, condominium are different concepts.

Done as suggested. The sample refers to the painted renderings applied over the inspected facades. Therefore, one case of a condominium of residences is tested, separated and with the rest of the sample, and the results showed that could be compared.

Table 1 and 2. Rendered facades, painted facades… The rendered facades are not painted? Table 3 brings together the 2 concepts Figure 2. The adjustment of the linear regression does not seem very enlightening and the margin of error is very significant. Is it possible to adjust better and draw more evident conclusions? Or be critical of the results obtained?

This question was clarified in the revised version of the manuscript. The linear regression is the one that provides the best goodness of fit to the sample analysed, considering the physical deterioration of the painted renderings analysed. In fact, the maintenance activities, as discussed in conclusions, usually are not performed uniformly, and only the cleaning actions allow drawing more evident conclusions.

There is self-citation by one of the authors of the article throughout the text. 12 references to “Silva et al.” were identified.

The references have been cited due to their relevance for the current study. To avoid a high level of self-citation, seven references from Silva have been removed in the revised version of the manuscript.

Round 2

Reviewer 2 Report

The authors revised the initial text introducing the main changes suggested in the review.